# GRK2-Mediated Crosstalk Between β-Adrenergic and Angiotensin II Receptors Enhances Adrenocortical Aldosterone Production In Vitro and In Vivo

**DOI:** 10.3390/ijms21020574

**Published:** 2020-01-16

**Authors:** Celina M. Pollard, Jennifer Ghandour, Natalie Cora, Arianna Perez, Barbara M. Parker, Victoria L. Desimine, Shelby L. Wertz, Janelle M. Pereyra, Krysten E. Ferraino, Jainika J. Patel, Anastasios Lymperopoulos

**Affiliations:** Laboratory for the Study of Neurohormonal Control of the Circulation, Department of Pharmaceutical Sciences (Pharmacology), College of Pharmacy, Nova Southeastern University, Fort Lauderdale, FL 33328, USA; cp1743@mynsu.nova.edu (C.M.P.); jg2901@mynsu.nova.edu (J.G.); nc1174@mynsu.nova.edu (N.C.); ap2491@mynsu.nova.edu (A.P.); barbaramparker@gmail.com (B.M.P.); vd359@mynsu.nova.edu (V.L.D.); sw1541@mynsu.nova.edu (S.L.W.); jm3706@yahoo.com (J.M.P.); kf713@mynsu.nova.edu (K.E.F.); jp2812@mynsu.nova.edu (J.J.P.)

**Keywords:** adrenocortical zona glomerulosa cell, aldosterone, angiotensin II, β-adrenergic receptor, G protein-coupled receptor-kinase-2, signaling crosstalk

## Abstract

Aldosterone is produced by adrenocortical zona glomerulosa (AZG) cells in response to angiotensin II (AngII) acting through its type I receptors (AT_1_Rs). AT_1_R is a G protein-coupled receptor (GPCR) that induces aldosterone via both G proteins and the adapter protein βarrestin1, which binds the receptor following its phosphorylation by GPCR-kinases (GRKs) to initiate G protein-independent signaling. β-adrenergic receptors (ARs) also induce aldosterone production in AZG cells. Herein, we investigated whether GRK2 or GRK5, the two major adrenal GRKs, is involved in the catecholaminergic regulation of AngII-dependent aldosterone production. In human AZG (H295R) cells in vitro, the βAR agonist isoproterenol significantly augmented both AngII-dependent aldosterone secretion and synthesis, as measured by the steroidogenic acute regulatory (StAR) protein and CYP11B2 (aldosterone synthase) mRNA inductions. Importantly, GRK2, but not GRK5, was indispensable for the βAR-mediated enhancement of aldosterone in response to AngII. Specifically, GRK2 inhibition with Cmpd101 abolished isoproterenol’s effects on AngII-induced aldosterone synthesis/secretion, whereas the GRK5 knockout via CRISPR/Cas9 had no effect. It is worth noting that these findings were confirmed in vivo, since rats overexpressing GRK2, but not GRK5, in their adrenals had elevated circulating aldosterone levels compared to the control animals. However, treatment with the β-blocker propranolol prevented hyperaldosteronism in the adrenal GRK2-overexpressing rats. In conclusion, GRK2 mediates a βAR-AT_1_R signaling crosstalk in the adrenal cortex leading to elevated aldosterone production. This suggests that adrenal GRK2 may be a molecular link connecting the sympathetic nervous and renin-angiotensin systems at the level of the adrenal cortex and that its inhibition might be therapeutically advantageous in hyperaldosteronism-related conditions.

## 1. Introduction

Aldosterone is an adrenocortical mineralocorticoid hormone with significant cardiovascular toxicity when produced in excess, as it contributes to hypertension, heart failure, and other heart conditions [1,2,3]. It is produced and secreted by the adrenal cortex, mainly in response to elevated serum potassium levels or angiotensin II (AngII) acting through AngII type I receptors (AT_1_Rs), which are endogenously expressed in adrenocortical zona glomerulosa (AZG) cells [4,5]. AT_1_R is a G protein-coupled receptor (GPCR) that also signals through G protein-independent pathways, a plethora of which are mediated by the scaffolding actions of βarrestins, which were originally discovered as terminators of GPCR signaling following receptor phosphorylation by a GPCR-kinase (GRK) [6]. In previous work, we have established that AT_1_Rs induce aldosterone synthesis and secretion via both a G_q/11_ protein/phospholipase C (PLC)-mediated signaling pathway and a βarrestin1-mediated pathway leading to sustained extracellular signal-regulated kinase (ERK) activation in AZG cells [7]. Subsequently, ERKs transcriptionally upregulate the steroidogenic acute regulatory (StAR) protein, the rate-limiting enzyme of aldosterone biosynthesis responsible for mitochondrial uptake of cholesterol, and the precursor of all adrenal steroids [8,9]. 

Catecholamines, such as the sympathetic nervous system hormones norepinephrine and epinephrine, are known to potentiate AngII actions in various tissues, including the adrenal cortex [10,11,12,13]. Indeed, AZG cells have been reported to endogenously express all three subtypes (β_1_, β_2_, β_3_) of the β-adrenergic receptor (AR) and to respond to catecholaminergic stimulation by producing and secreting aldosterone and other adrenal steroid hormones [14]. Being GPCRs, βARs also activate GRKs and βarrestins, like the AT_1_R does, to signal to downstream effectors [15,16,17,18]. In an effort to delineate the signaling mechanisms that underlie the catecholaminergic modulation of AngII-dependent aldosterone production in the adrenal cortex, we investigated, in the present study, the effects of βAR stimulation on AngII-dependent aldosterone production in the human AZG cell line H295R, as well as the potential involvement of the most abundant GRK isoforms in the adrenal gland, GRK2 and GRK5 [19], in this process. 

## 2. Results

### 2.1. Catecholamines and AngII Synergistically Stimulate Aldosterone Secretion from H295R Cells

In a first set of experiments, we investigated whether βARs can modulate AT_1_R-induced aldosterone secretion from H295R AZG cells in vitro. As expected, either βAR stimulation with the non-subtype selective full agonist isoproterenol (10 µM) or AT_1_R stimulation with AngII (100 nM) alone induced aldosterone secretion (~2-fold and ~3-fold of no stimulation, respectively) (Figure 1). Interestingly, the simultaneous application of both 10 µM isoproterenol and 100 nM AngII led to a significantly increased aldosterone secretion, more than doubling the response to either agent alone (~7-fold of no stimulation) (Figure 1). This suggests that βARs and AT_1_Rs, i.e., catecholamines and AngII, exert a synergistic effect on aldosterone secretion stimulation in AZG cells. The pretreatment with the MAPK/ERK kinase (MEK)-1 inhibitor PD98059, which blocks ERK activation, at a standard, non-toxic level for the cells concentration (50 µM) [7], abolished aldosterone secretion in response to either isoproterenol or AngII (Figure 1), indicating that ERKs are necessary for stimulating aldosterone production (and secretion) via either βARs or AT_1_Rs (or both), or, most probably, via downstream StAR gene induction [7,8].

### 2.2. GRK2, but Not GRK5, Is Essential for the Synergism between Catecholamines and AngII to Stimulate Aldosterone Production

Since both βAR and AT_1_R can activate the essential for aldosterone synthesis ERKs by interacting with βarrestins in a GRK phosphorylation-dependent manner [20], we next investigated the roles of GRK2 and GRK5, the most abundant adrenal GRKs [19], in the βAR-AT_1_R crosstalk during the stimulation of aldosterone production. As shown in Figure 2A, neither pharmacological GRK2 blockade with Cmpd101 [21], nor GRK5 CRISPR-mediated knockout (KO) (Figure 2B) alone could affect isoproterenol- or AngII-induced aldosterone secretion in a statistically significant manner. Importantly, vehicle (DMSO) alone and Cmpd101 alone were applied to mock CRISPR lentivirus-infected cells and had no effect on aldosterone secretion (data not shown). However, the combination of the GRK2 blockade and GRK5 genetic deletion significantly reduced (albeit not completely abolished) isoproterenol- and AngII-induced aldosterone secretion (Figure 2A). In contrast, GRK2 blockade with Cmpd101 alone, but not GRK5 genetic deletion alone, was sufficient to completely abolish the synergistic effect of the combined isoproterenol and AngII application on aldosterone secretion in H295R cells (Figure 2A). This suggests that GRK2, but not GRK5, is responsible for the synergistic crosstalk between βAR and AT_1_R during the stimulation of aldosterone production in AZG cells. The combined GRK2 blockade and GRK5 KO, again, significantly reduced, but did not completely abolish, the isoproterenol + AngII-induced aldosterone secretion (Figure 2A).

### 2.3. GRK2, but not GRK5, Is Essential for the Synergistic Aldosterone Synthesis Induction By Catecholamines and AngII In AZG Cells

To further corroborate the essential role of GRK2 in the uncovered βAR-AT_1_R crosstalk during aldosterone induction in AZG cells, we also checked for the effects of GRK2 and GRK5 in stimulation of StAR and of CYP11B2 (aldosterone synthase) gene expressions (i.e., mRNA inductions) by the combined isoproterenol + AngII treatment. Consistent with the in vitro aldosterone secretion experiments above (Figure 2), real-time quantitative PCR revealed that GRK2 blockade with Cmpd101 abolished the isoproterenol-dependent increase in AngII-induced StAR mRNA levels, whereas GRK5 KO had no effect (Figure 3A). The same held true for CYP11B2 mRNA induction (Figure 3B). Combining GRK2 pharmacological blockade and GRK5 deletion further reduced isoproterenol + AngII-stimulated StAR (Figure 3A) and CYP11B2 (Figure 3B) mRNA levels below those observed with AngII treatment alone, i.e., it caused a bigger reduction than that achieved with GRK2 inhibition alone. Notably, these treatments were performed also in mock CRISPR lentivirus-infected cells with identical results as in the “-GRK5 KO” cells (first three, from the left, bars in Figure 3A and first five, from the left, bars in Figure 3B), indicating the absence of non-specific effects by the CRIPSR virus transfections. Taken together, these results strongly suggest that GRK2, and not GRK5, is necessary for the combined effect of βAR and AT_1_R activation on StAR and CYP11B2 gene inductions, and hence, on aldosterone synthesis, in AZG cells.

### 2.4. βAR-Activated GRK2, but Not GRK5, Is Essential for the Catecholamine-Dependent Enhancement of Adrenal Aldosterone Production In Vivo

In order to confirm the validity of our in vitro findings from the aforementioned H295R cells in vivo, we overexpressed GRK2 or GRK5 specifically in the adrenals of normal, adult, healthy rats, via a direct intra-adrenal injection of recombinant adenoviruses, encoding for wild type, full length versions of these two GRKs, as we have done previously [7,22]. After confirming adrenal-specific GRK transgene overexpression (Figure 4A), we measured circulating aldosterone levels in these animals at 7 days post-gene delivery and compared them to those of the control rats, whose adrenals were injected with green fluorescent protein (GFP)-encoding control adenovirus (the entire flowchart of the experiment is shown in Figure 4B). As shown in Figure 4C, the adrenal GRK2-overexpressing rats displayed significant hyperaldosteronism compared to control adrenal GFP-expressing rats. In contrast, plasma aldosterone levels in adrenal GRK5-overexpressing rats were indistinguishable from the controls (Figure 4C). This strongly indicates that, similar to cultured AZG cells in vitro, GRK2, but not GRK5, promotes adrenocortical aldosterone production in vivo. In addition, this effect of adrenal GRK2 was βAR-dependent, since circulating aldosterone levels in adrenal GRK2-overexpressing rats that were simultaneously administered the βAR antagonist propranolol (starting on the day of the GRK2 adrenal gene delivery) were not higher than in the control (AdGFP) animals, with or without concomitant propranolol treatment (Figure 4C). It is worth noting that propranolol had no effect on circulating aldosterone levels in control (AdGFP) rats (Figure 4C), which probably reflects the fact that AngII, rather than catecholamines, is the primary endogenous hormonal stimulus for adrenal aldosterone production in vivo [23]. Finally, adrenal GRK5 overexpression did not affect blood aldosterone levels, regardless of the β-blocker treatment (Figure 4C), which suggests that this GRK is not substantially involved in the regulation of adrenal aldosterone production in vivo.

## 3. Discussion

In the present study, we have uncovered a previously unappreciated role for GRK2 (Figure 5) in mediating a signaling crosstalk between βARs and AT_1_Rs in AZG cells in vitro and in vivo. This GRK2-dependent crosstalk between these two different GPCR types markedly enhances the synthesis and secretion of aldosterone induced by either AngII or endogenous catecholamine signal alone in AZG cells (Figure 5). This crosstalk represents one of the various molecular/signaling mechanisms by which the sympathetic nervous system trans-regulates the renin-angiotensin-aldosterone system (RAAS) at the level of adrenocortical aldosterone synthesis and secretion. Thus, in situations of acute stress, where circulating catecholamine levels are elevated, the mechanism uncovered here ensures that circulating aldosterone levels also increase to help the cardiovascular, immune, and the myriad of other organ systems that aldosterone affects with its actions prepare their responses to the stressful insult [10,24]. On the other hand, in chronic conditions characterized by elevated sympathetic activity (e.g., chronic heart failure), this mechanism would contribute further to the morbidity and mortality of the disease by causing hyperaldosteronism, a maladaptive and confounding feature/symptom of almost every pathological condition in which it occurs [1,10]. Indeed, in chronic heart failure in particular, adrenal GRK2 upregulation has already been established as an absolutely essential process for the induction and maintenance of the chronic sympathetic nervous system hyperactivity that accompanies and aggravates this disease thanks to the GRK2-dependent termination/blockade of the sympatho-inhibitory α_2_AR signaling in the chromaffin cells of the adrenal medulla [19,22,25,26]. Based on our present findings, adrenal GRK2 upregulation could also very well lead to the elevated circulating aldosterone levels that are likewise known to accompany and complicate human chronic heart failure [2,3,5,17,27]. This possibility warrants future investigation in vivo in animal heart failure models. 

As far as the exact molecular mechanism of GRK2 involvement in this adrenocortical βAR-AT_1_R crosstalk, there are several possibilities worth exploring in future studies. One of them is that, while GRK5 is permanently anchored to the plasma membrane, GRK2 is cytoplasmic, at rest, and requires an interaction with the free G_βγ_ subunits of activated G proteins for its membrane tanslocation [28]. Therefore, GRK2 may require activation by another GPCR (e.g., βAR in this case) in order to phosphorylate the AT_1_R in AZG cells, whereas GRK5, being constitutively membrane-anchored, lacks such a requirement. Another possibility is the involvement of a yet-to-be unidentified protein in close proximity with the AT_1_R that preferentially recruits GRK2 (over GRK5) in AZG cell membranes. Interestingly, the AT_1_R was recently shown to heterodimerize with the β_2_AR, enhancing βarrestin2 interaction with the latter receptor in transfected HEK293 cells [29]. Although no GRK involvement in this was examined, this study raises the intriguing possibility that AT_1_Rs and βARs form heterodimers also in AZG cells and that these heterodimers then become substrates for GRK2 only, enhancing βarrestin1 recruitment and downstream signaling to aldosterone production. Finally, the herein reported GRK2-dependent βAR-AT_1_R crosstalk in AZG cells might not take place at the level of the receptors per se and the signaling of the two receptors could converge at some receptor-proximal downstream effector modulated by GRK2 instead. For instance, the calcium-regulated chloride channel TMEM16A (anoctamine-1), which is activated by calcium/calmodulin-dependent kinase phosphorylation, was very recently implicated in the stimulation of aldosterone production in AZG cells [30,31]. βAR-activated GRK2 may very well phosphorylate and activate this channel, thereby mediating the stimulation of aldosterone secretion by both βARs and AT_1_Rs (AT_1_R elicits intracellular [Ca^2+^] elevation via G_q/11_ proteins [23], and Ca^2+^ activates TMEM16A). An investigation of all the possible mechanisms mentioned above that may underlie this crosstalk are under way in our laboratory.

In summary, we report here that βAR-stimulated GRK2 mediates an important signaling crosstalk between βARs and AT_1_Rs in AZG cells, resulting in the catecholaminergic augmentation of AngII-dependent aldosterone synthesis and secretion. Importantly, this finding holds true also in vivo in the adrenal cortex, and this property of GRK2 is not shared by the other major adrenal GRK isoform, GRK5. Therefore, adrenal GRK2 could be a key molecular and biochemical link connecting the adrenergic nervous system with the RAAS at the level of adrenocortical aldosterone production. Additionally, GRK2 is selectively upregulated in the adrenal gland (and in sympathetic neurons) in chronic heart failure [10], a disease characterized by both sympathetic nervous system hyperactivity and hyperaldosteronism [1,10]. This means that adrenal GRK2 upregulation could very well be the molecular culprit behind the increases in the circulating levels of both catecholamines and aldosterone in human chronic heart failure. Thus, adrenal-specific GRK2 pharmacological inhibition could confer a double benefit for the failing heart in terms of cardio-toxic hormone burden relief.

## 4. Materials and Methods

### 4.1. Materials

Cmpd101 was from HelloBio (Cat. #HB2840, Bristol, UK; >99% purity, as assessed by HPLC). All other chemicals (isoproterenol-Cat. #I6504, AngII-Cat. #A9525, PD98059-Cat. #P215, propranolol-Cat. #P0884) were from Sigma–Aldrich (St. Louis, MO, USA; ≥98% purity, as assessed by HPLC).

### 4.2. H295R Cell Culture and Transfections

H295R cells were purchased from the American Type Culture Collection (Manassas, VA, USA; RRID: CVCL_0458) and cultured as previously described [7,32,33]. Cells used for all experiments were not passaged more than 3 times (passage ≤ 3). Recombinant lentiviruses encoding for the rat wild type full-length GRK5 or for the empty vector (control) were purchased from OriGene Technologies (Rockville, MD, USA) and propagated in HEK293 cells, followed by two sequential rounds of CsCl density gradient ultracentrifugation for purification [7,19]. For CRISPR/Cas9-mediated gene deletion, a human GRK5 gene-specific gRNA sequence was custom-designed and synthesized by Sigma–Aldrich and then inserted into Sigma’s proprietary lentiviral backbone vector (U6-gRNA:ef1a-puro-2A-Cas9-2A-tGFP) for lentiviral particle production. The resultant viral particles were purchased from Sigma, along with the negative control CRISPR lentiviral particles (CNCV, to be used as a control for lentiviral-mediated CRISPR/Cas9 gene deletion; Cat #CRISPR12V-1EA, which is a lentiviral system containing a gRNA sequence that does not target known human, mouse, or rat genes). Both lentiviruses were propagated in HEK293 cells and purified through two sequential rounds of CsCl density gradient ultracentrifugation. 2 × 10^6^ purified viral particles per 1 × 10^6^ cells in culture were generally used for cell infection. Forty-eight hours after infection, protein extracts were prepared from the cells to confirm the absence of GRK5 protein via Western Blotting (see below). Cells from the same pool that was infected with the GRK5-specific CRISPR lentivirus and was confirmed to be GRK5-depleted were used for drug treatments that same day (i.e., 48 h post-infection). 

### 4.3. Aldosterone Measurements

In vitro aldosterone secretion in the culture medium of H295R cells and aldosterone levels in rat blood serum were measured by enzyme immune-assay (EIA) (Aldosterone EIA kit, Cat. #: 11-AD2HU-E01; ALPCO Diagnostics, Salem, NH, USA), as described [7,32,34].

### 4.4. Real-Time PCR

Total RNA isolation with TRIzol reagent (Life Technologies, Grand Island, NY, USA), reverse transcription and real-time quantitative RT-PCR analysis were carried out as previously described [17,19]. The following primer pairs were used: 5′-GGCATCCTTAGCAACCAAGA-3′ and 5′-TCTCCTTGACATTGGGGTTC-3′ for human StAR; 5′-CCTTGCTGATGACGCTCTTTG-3′ and 5′-CTCACCACTCGCTCCAAAAACA-3′ for human CYP11B2; 5′-TCAAGAACGAAAGTCGGAGG-3′ and 5′-GGACATCTAAGGGCATCAC-3′ for 18S rRNA. A real-time PCR analysis was performed using a SYBR^®^ Green Supermix (Bio-Rad Laboratories, Hercules, CA, USA). The normalization was done with the housekeeping gene 18S rRNA levels. No bands were seen in the absence of reverse transcriptase (data not shown).

### 4.5. Experimental Animals and Gene Delivery Procedure

All animal procedures and experiments were performed in accordance with the guidelines of the IACUC committee of Nova Southeastern University. All animals used in the study were handled and sacrificed in compliance with the American Veterinary Medical Association’s (AVMA) guidelines on animal handling and euthanasia. Adrenal-specific in vivo gene delivery in 250–300 g, 2-month-old male Sprague-Dawley rats was done as described [22] (via the direct injection of adenovirus in both adrenal glands). The constructs used for AdGRK2 and AdGRK5 production and propagation have been described previously [7,19]. For intra-adrenal injections, 1.3 × 10^10^ total particles, diluted in 100 μL phosphate-buffered saline (PBS), of each type of adenovirus were rapidly injected via a 31-gauge needle.

### 4.6. Western Blotting

H295R cell and rat adrenal protein extracts were prepared, as previously described [7,32,34], in a 20 mM Tris pH 7.4 buffer containing 1% Nonidet P-40, 20% glycerol, 10 mM PMSF, 1 mM Na_3_VO_4_, 10 mM NaF, 2.5 µg/mL aprotinin, and 2.5 µg/mL leupeptin. The protein concentration was determined via the bicinchoninic acid (BCA) method and equal amounts of protein per sample were loaded. The following antibodies were used for immunoblotting: sc-565 (Santa Cruz Biotechnology, Santa Cruz, CA, USA; RRID: AB_2115456) for GRK5; sc-8329 (Santa Cruz Biotechnology; RRID: AB_647326) for GRK2; and sc-47724 (Santa Cruz Biotechnology; RRID: AB_627678) for GAPDH. Immunoblots were revealed by enhanced chemiluminescence (ECL, Life Technologies, Grand Island, NY, USA) and visualized in the FluorChem E Digital Darkroom (Protein Simple, San Jose, CA, USA), as described previously [15,33,35]. Densitometry was performed with the AlphaView software (Protein Simple) in the linear range of signal detection (on non-saturated bands).

### 4.7. Statistical Analyses

The data are generally expressed as mean ± SEM. An unpaired 2-tailed Student’s *t* test and a one- or two-way ANOVA with Bonferroni test were generally performed for statistical comparisons using the SPSS 23 software (SPSS, Inc., Chicago, IL, USA). For all tests, a *p* value of <0.05 was generally considered to be significant.

## Figures and Tables

**Figure 1 ijms-21-00574-f001:**
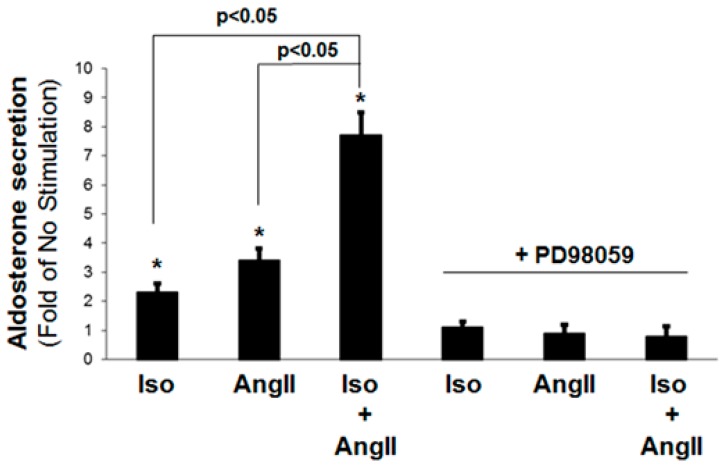
Synergistic effect of catecholamines and AngII on stimulation of aldosterone secretion from H295R cells. Aldosterone secretion in vitro by H295R cells stimulated for 6 h with 10 µM isoproterenol alone (Iso), 100 nM AngII alone (AngII), or both (applied simultaneously) (Iso + AngII). The results of each of these treatments in the presence of 50 µM PD98059 are also shown. The data are expressed as fold of the response to vehicle (No Stimulation). *, *p* < 0.05, vs. no stimulation (vehicle); *n* = 5 independent experiments/treatment.

**Figure 2 ijms-21-00574-f002:**
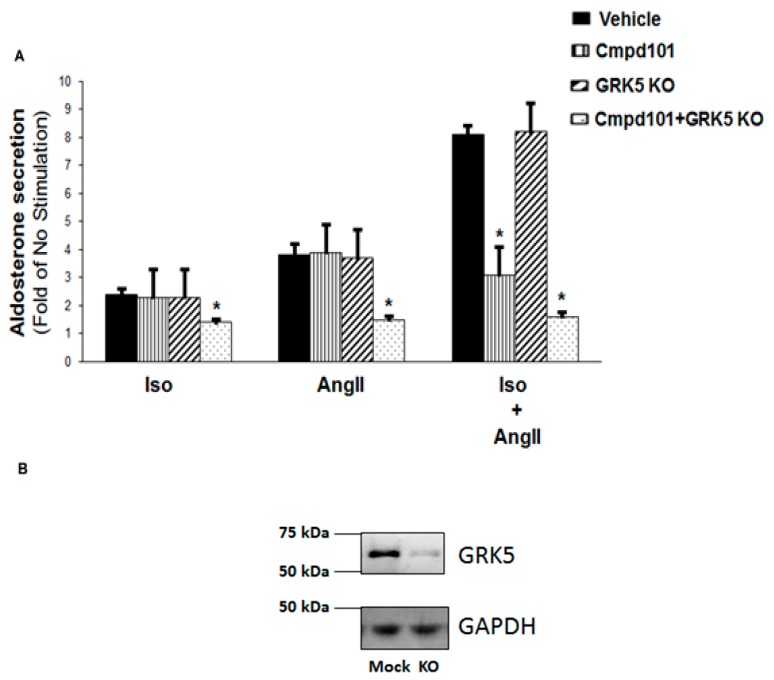
GRK2 mediates the synergism between βARs and AT_1_Rs in adrenocortical zona glomerulosa (AZG) cells, leading to enhanced aldosterone production. (**A**) Aldosterone secretion was measured at 6 hr post-challenge, with 10 µM isoproterenol alone (Iso), 100 nM AngII alone (AngII), or both (applied simultaneously) (Iso + AngII) from control (no manipulation-Vehicle) H295R cells, from cells pretreated with 30 µM Cmpd101, from cells having GRK5 genetically deleted via CRISPR (GRK5 KO), or from cells having both GRK5 genetically deleted and pretreated with 30 µM Cmpd101 (Cmpd101 + GRK5 KO). The data are expressed as a fold of the response to no stimulation. *, *p* < 0.05, vs. corresponding Vehicle; *n* = 5 independent experiments/treatment. (**B**) Immunoblotting for GRK5 in extracts from cultured H295R cells, transfected with control empty vector/mock lentivirus (Mock) or CRISPR human GRK5-specific lentivirus to delete the gene for GRK5 (KO). A representative blot is shown, including glyceraldehyde 3-phosphate dehydrogenase (GAPDH) as the loading control of three independent experiments performed in duplicate, confirming GRK5 deletion in KO cells.

**Figure 3 ijms-21-00574-f003:**
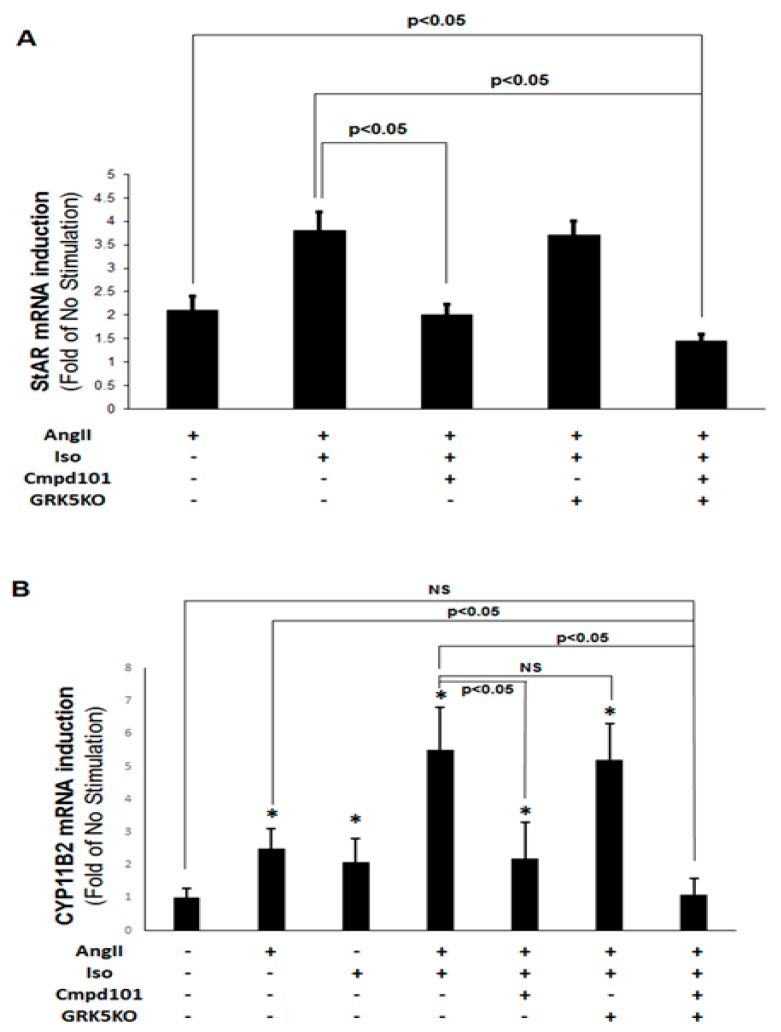
GRK2 mediates the βAR-induced enhancement of AngII-dependent aldosterone synthesis in AZG cells. H295R cells were treated for 2 h with 100 nM AngII alone or with 100 nM AngII and 10 µM isoproterenol (Iso) combined, in the presence of 30 µM Cmpd101 or after transfection with a CRIPSR lentivirus to knockout GRK5 (GRK5KO) or both (Cmpd101 + GRK5KO). At the end of the 2-h-long treatment, cells were harvested and total RNA was isolated to perform real-time PCR for StAR (**A**) or CYP11B2 (**B**) mRNA levels quantitation. Results were normalized based on the mRNA levels of the housekeeping gene 18S and are expressed as fold of control (no stimulation). (**A**) *n* = 5 independent experiments/group; (**B**) *, *p* < 0.05, vs. Control (No stimulation); NS: Not significant at *p* = 0.05; *n* = 5 independent experiments/group.

**Figure 4 ijms-21-00574-f004:**
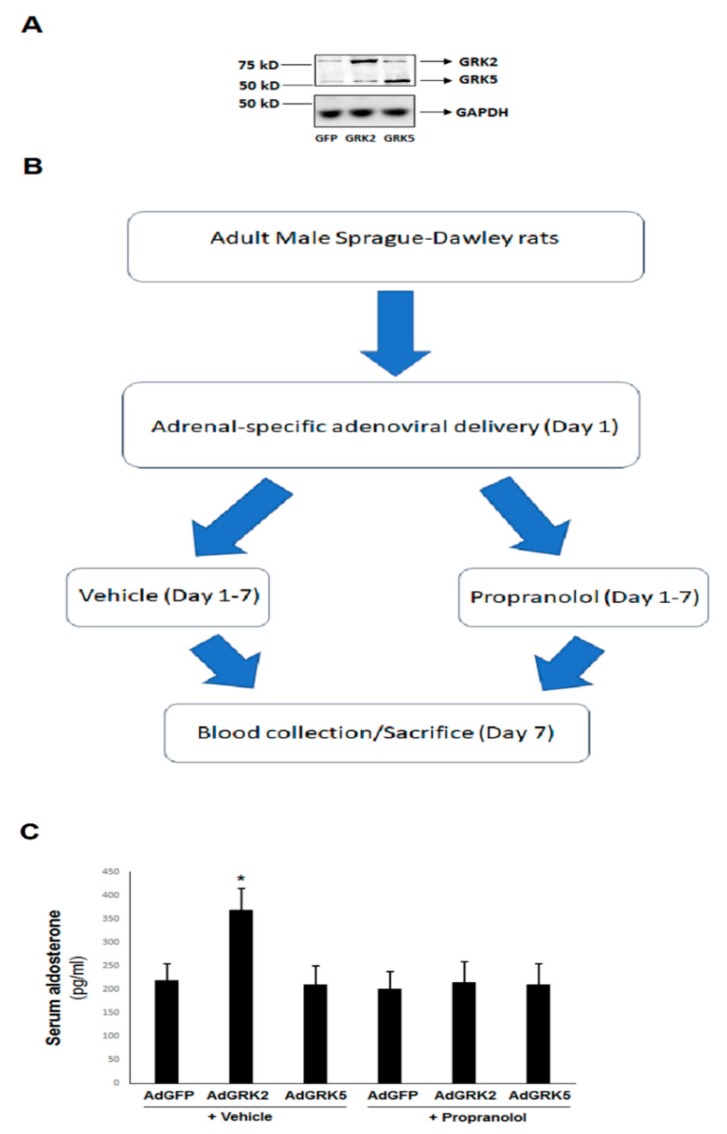
βAR-induced GRK2, but not GRK5, is essential for adrenal aldosterone production in vivo. (**A**) Representative western blots in protein extracts from the adrenal glands from AdGFP (GFP)-, AdGRK2 (GRK2)-, or AdGRK5 (GRK5)-treated healthy adult rats, at 7 days post in vivo gene transfer, confirm the overexpression of the respective transgenes. Glyceraldehyde 3-phosphate dehydrogenase (GAPDH) is also shown as a loading control. (**B**) A flowchart of the in vivo procedural sequence. (**C**) The circulating aldosterone levels in the rats treated with 100 mg/kg/day propranolol or vehicle (in drinking water) for seven consecutive days. *, *p* < 0.05, vs. any other group; *n* = 5 rats/group.

**Figure 5 ijms-21-00574-f005:**
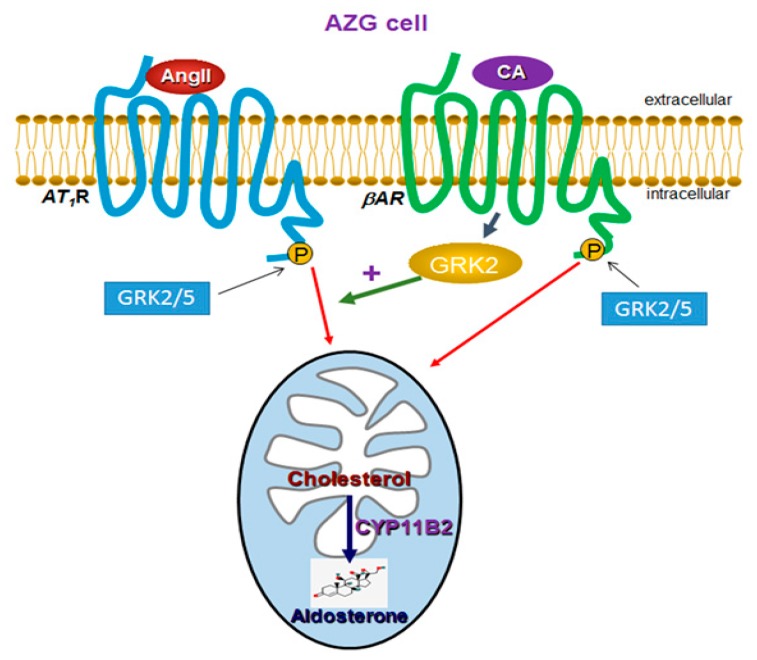
Schematic illustration of the GRK2-dependent signaling crosstalk between βARs and AT_1_Rs in AZG cells leading to enhanced aldosterone production in vitro and in vivo. Although both GRK2 and -5 can phosphorylate either receptor to induce downstream signaling resulting in aldosterone synthesis in the mitochondria of the AZG cell, βAR-activated GRK2 (but not GRK5) augments the AT_1_R signal to aldosterone induction in AZG cells. CA: Catecholamine. See text for more details and for all other molecular acronym definitions.

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
