# Peer review of "GRK2-Mediated Crosstalk Between β-Adrenergic and Angiotensin II Receptors Enhances Adrenocortical Aldosterone Production In Vitro and In Vivo"

_ijms, 2020, doi:10.3390/ijms21020574_

Round 1
Reviewer 1 Report
The manuscript submitted for review concerns the role of GRK2 or GRK5, the two major adrenal GRKs, in catecholaminergic regulation of AngII-dependent aldosterone production. The publication contains a series of original results obtained via novel molecular biology methods. After minor corrections I recommend it for publication.
Remarks
1 The materials and methods described in a rather superficial way make it impossible to reproduce the described procedures. They often refer to previous publications - please refer to publications only in which the procedure was described. Please avoid the queues of references to a given procedure. For example, authors write:
"vitro aldosterone secretion in the culture medium of H295R cells and aldosterone levels in rat blood serum were measured by EIA (Aldosterone EIA kit, Cat. #: 11- AD2HU-E01; ALPCO Diagnostics, Salem, NH, USA), as described [7,20,21].
"
However, in publication no. 7 to which they refer there is an identical entry referring to another publication: "Rat plasma aldosterone levels and in vitro aldosterone secretion in the culture medium of H295R cells were determined by EIA (Aldosterone EIA kit; ALPCO Diagnostics), as described (34)"
2 No data concerning age, sex of animals used in vivo experiments
3 No data on the size of individual groups
4 The authors used the CRISPR method - which is quite a challenge in the context of H295R cells that tend to spontaneously polyploidy. Please describe the procedure in more detail and provide the passage number of the used cells.
Author Response
Remarks
1 The materials and methods described in a rather superficial way make it impossible to reproduce the described procedures. They often refer to previous publications - please refer to publications only in which the procedure was described. Please avoid the queues of references to a given procedure. For example, authors write:
"vitro aldosterone secretion in the culture medium of H295R cells and aldosterone levels in rat blood serum were measured by EIA (Aldosterone EIA kit, Cat. #: 11- AD2HU-E01; ALPCO Diagnostics, Salem, NH, USA), as described [7,20,21].
"
However, in publication no. 7 to which they refer there is an identical entry referring to another publication: "Rat plasma aldosterone levels and in vitro aldosterone secretion in the culture medium of H295R cells were determined by EIA (Aldosterone EIA kit; ALPCO Diagnostics), as described (34)"
Author response: We thank this reviewer for their overall kind and positive comments on the quality of our work. We are not exactly sure what the reviewer`s concern is here but, in any case, we have expanded our "Materials and Methods" section substantially in the revised manuscript (additions highlighted in yellow), as per his/her request. We hope this satisfies now this reviewer.
2 No data concerning age, sex of animals used in vivo experiments
Author response: Added in revised "Materials and Methods" section.
3 No data on the size of individual groups
Author response: This is available in Figure 4`s legend. As stated therein, randomized animal groups were 5 rats/adenoviral treatment/drug treatment.
4 The authors used the CRISPR method - which is quite a challenge in the context of H295R cells that tend to spontaneously polyploidy. Please describe the procedure in more detail and provide the passage number of the used cells.
Author response: This is a very valid point raised by the reviewer. We could not agree more that the CRISPR method is quite challenging and poses an arduous task for achieving genetic ablation. It was no wonder that it took us more than 6 months of continuous trial-and-error experiments to standardize the method in our laboratory! Nonetheless, we describe the procedure and the specific lentiviral constructs used in significantly more detail now in the revised "Materials and Methods" section, as per the reviewer`s request. As for the passage of the cells used in the present study, it never exceeded 3. We clearly state this also now in the "Materials and Methods" section of the revised manuscript. We hope this now satisfies this reviewer.
Reviewer 2 Report
The manuscript presents an interesting study that investigated in vitro and then confirms in vivo on a murine model the implication of βAR-stimulated GRK2 in catecholaminergic augmentation of AngII-dependent aldosterone synthesis and secretion. These findings are very important for the personalized treatment of heart failing in which GRK2 is upregulated and can be the main cause of sympathetic nervous system hyperactivity and hyperaldosteronism.
The manuscript is well written, but some aspects can be improved especially in the material and methods part.
In the material and methods part is not described the experimental part, the treatment of the H295R cells, it will be more explicit to introduce a flowchart of the experiment design. For the animal experiment should also be added a description and a flow chart for more understanding.
Author Response
1) The manuscript is well written, but some aspects can be improved especially in the material and methods part.
Author response: We thank this reviewer for the overall kind and positive comments about the quality of our work. In accordance also with Reviewer #1`s similar comment/suggestion, we have now substantially expanded the "Materials and Methods" section of our revised manuscript in much more experimental detail (all additional information is highlighted in yellow).
2) In the material and methods part is not described the experimental part, the treatment of the H295R cells, it will be more explicit to introduce a flowchart of the experiment design. For the animal experiment should also be added a description and a flow chart for more understanding.
Author response: Again, similarly to the preceding comment, we now describe the experimental part pertaining to the H295R cells in more detail in the revised "Materials and Methods" section. We have also added a flowchart of the in vivo animal experiment, as a new "Figure 4B" panel, in the revised manuscript, in accordance with the reviewer`s excellent suggestion. We hope this now satisfies this reviewer.